# Pressure Point Thresholds and ME/CFS Comorbidity as Indicators of Patient’s Response to Manual Physiotherapy in Fibromyalgia

**DOI:** 10.3390/ijerph17218044

**Published:** 2020-10-31

**Authors:** Francisco Javier Falaguera-Vera, María Garcia-Escudero, Javier Bonastre-Férez, Mario Zacarés, Elisa Oltra

**Affiliations:** 1Escuela de Doctorado, Universidad Católica de Valencia San Vicente Mártir, 46008 Valencia, Spain; fj.falaguera@ucv.es (F.J.F.-V.); javier.bonastre@mail.ucv.es (J.B.-F.); 2School of Health Sciences, Universidad Católica de Valencia San Vicente Mártir, 46900 Valencia, Spain; maria.escudero@ucv.es; 3Centro de Investigación Traslacional San Alberto Magno, Universidad Católica de Valencia San Vicente Mártir, 46001 Valencia, Spain; 4School of Experimental Sciences, Universidad Católica de Valencia San Vicente Mártir, 46001 Valencia, Spain; mario.zacares@ucv.es; 5School of Medicine, Universidad Católica de Valencia San Vicente Mártir, 46001 Valencia, Spain

**Keywords:** fibromyalgia, myalgic encephalomyelitis/chronic fatigue syndrome (ME/CFS), pressure point threshold (PPT), physiotherapy, manual therapy (MT)

## Abstract

Current pharmacological treatments of Fibromyalgia (FM) are merely symptom palliative, as clinical trials have so far failed to provide overall benefits without associated harms. Polypharmacy often leads to patient’s health deterioration and chronic drug use to an eventual lack of patient’s response. Emerging evidence supports that physiotherapy treatments based on mechanical triggers improve FM symptoms and therefore could be used for therapeutic purposes by themselves or in combination with current pharmacological treatments, as part of integrative medicine programs. However, a paucity of studies rigorously and systematically evaluating this possibility exists. This study uses scores from validated standardized questionnaires, algometer pressure point threshold (PPT) readings and responses from a custom self-developed questionnaire to determine the impact of a pressure-controlled custom manual protocol on FM hyperalgesia/allodynia, fatigue and patient’s quality of life. The results show that patient’s baseline sensitivity to pain inversely correlates with treatment response in FM. Moreover, post-stratification analysis unexpectedly reveals that patients presenting comorbid ME/CFS do not seem to respond to the applied therapy as those presenting FM only. Therefore, pre-treatment PPTs and ME/CFS comorbidity may serve as indicators to predict patient’s response to physiotherapy programs based on mechanical triggers. Further exploration of these findings is granted. In addition, the study of gene expression profiles in the blood collection generated by this study should help unveil the molecular mechanisms behind patient’s differential response to manual therapy.

## 1. Introduction

The tenth revision of the International Classification of Diseases, Clinical Modification (ICD-10-CM), assigns the M79.7 to Fibromyalgia (FM) and defines the disease, together with fibromyositis, fibrositis and myofibrositis, as a chronic multisymptomatic disease of unknown etiology. FM presents with low pain threshold, muscle stiffness and tenderness, often accompanied with sleep disturbances, general fatigue, headaches and memory loss [1,2,3,4]. Similarly, Myalgic Encephalomyelitis/Chronic Fatigue Syndrome (ME/CFS) was classified with the code R53.82, or G93.3 if post-viral. The disease is described as an acquired, complex syndrome characterized by exercise-induced exacerbated fatigue, also known as PEM (post-exertional malaise), flu-like symptoms, sleep disturbances, cognitive dysfunction and orthostatic intolerance, in conjunction with others [5,6]. Some authors have posited that these two diseases are part of the same somatic syndrome, based on overlapping symptomatology [7,8,9]; some analytical coincidences like the increased lactate levels in ventricular cerebrospinal fluid, reported by B. Natelson et al. [10], and epidemiologic overlaps [11]. However, a list of differences in clinical and biological parameters, including PEM and autonomic function; hormone unbalance and differential cytokine, microRNA and gene expression profiles [12,13,14,15,16,17,18,19], suggests that the underlying pathophysiology in FM differs from that of ME/CFS [10]. 

Current pharmacological treatments for FM and/or CFS/ME patients are exclusively directed to palliate symptoms, since clinical trials have so far failed to provide overall benefits without associated harms [20,21,22]. In addition, polypharmacy often leads to patient deterioration, particularly for those suffering from chemical sensitivity, and extended drug use translates into patient desensitization and lack of drug-response. Non-pharmacological options are mainly dealing with CBT/GET (cognitive behavioral therapy by itself or combined with gradual exercise therapy). FM patients can engage in moderate to vigorous exercise; however, they often experience difficulties performing and adhering to even moderate intensity regimes because of increased pain symptoms [23]. On another side, the PACE trials design and outcomes continue under debate [24,25].

Physiotherapy-based treatments, manual therapy (MT) included, might provide exercise-like effects on treated tissues, by inducing blood flow and/or increased muscle tone, without demanding physical efforts from the patient and therefore, contrary to GET, should not compromise patient’s health. 

MT involves the manipulation of soft tissues and joints to relieve pain, reduce inflammation, release muscle contractures and increase the range of motion, to restore health. Massage maneuvers include pressures, rubbing, friction, kneading and vibrations, among others. Although different benefits are attributed to particular maneuvers, i.e., increased vagal tone by moderate pressures [26,27], and the duration, frequency, treated areas, repetition and pressures are known to be key on the downstream effects, to date, MT protocols are still poorly defined. 

Despite this lack of protocol rigor, a recent systematic review and a meta-analysis of RCTs (randomized clinical trials) show that MT improves pain, anxiety and depression in FM patients [28,29], suggesting that physiotherapy treatments based on manual maneuvers could be used for therapeutic purposes by themselves or in combination with current symptomatic pharmacological treatments, as part of integrative medicine programs. Inspired by this idea, our group reviewed clinical and research evidence of MT-based therapeutics with a focus on the role of the differential effects of pressure-therapeutics [30], finding that medium load pressure massage (4.5 *N*), using a particular frequency and repetitions as described by Dupont-Vergesteegden´s group [31], can be used as an intervention to aid in the regrowth of muscle lost during immobilization and in sedentary deconditioned individuals, such as severely affected FM and/or CFS/ME patients. At the same time, and similarly to CBT and mindfulness, MT might engage patient’s mind into relaxation, boosting happiness and perhaps immune, hormonal and neurotransmitter responses [24,32,33,34,35]. 

Documented with results from the clinic and those obtained from animal models and mimetic devices [30], our group developed a custom pressure-controlled MT protocol for the treatment of FM and evaluated patient’s response to this therapeutic program in the context of the registered Clinical Trial (CT) NCT04174300. Outcome measures included the use of standardized instruments, pressure-point threshold (PPT) digital recordings and scores from a custom questionnaire (CQ) created towards gathering patient’s self-impressions of the treatment. The aim of this study was to assess the benefits of our pressure-controlled MT protocol into alleviating FM symptoms according to questionnaire scores of the standardized instruments: FIQ (Fibromyalgia Impact Questionnaire) [36,37], MFI (Multi-Fatigue Inventory) [38], SF-36 [39] and our CQ. In addition, improvement of patient hyperalgesia/allodynia was determined by increased PPTs in the 18 FM tenderpoints described by the ACR1990 FM diagnostic criteria [2].

Furthermore, since we are interested in identifying the subjacent molecular mechanisms involved in MT mechano-transduction associated therapeutic effects, a biophysical process by which cells are capable of sensing their environment and of translating cues into biochemical signals [40,41], we used this CT as a platform to generate a collection of blood fractions for future gene expression inquiries, as depicted here.

## 2. Experimental Procedures

### 2.1. Study Design and Patient Recruitment

The study consisted in a pilot interventional non-randomized single arm CT, registered with the ClinicalTrials.gov Identifier: NCT04174300, for the treatment of FM symptoms. The program included 8 sessions of MT (twice weekly on alternate days) (Figure 1). Each session consisting in a 25-min custom protocol including pressure maneuvers of about 4.5 N, performed by a single operator (collegiate physiotherapist). Patient follow-up continued one month after treatment withdrawal. Participating patients (*n* = 40) were recruited by invitation through local patient associations. All participants fulfilled the ACR (American College of Rheumatology) 1990 and/or 2010 criteria [2,3] as referred by their local Medical Center. Comorbid ME/CFS diagnosis in compliance with Canadian and/or International ME/CFS criteria [5,6] was requested. Patient complete blood analytics within one month prior to study start were made available. Only patients not receiving hormonal therapy or suffering of previous serious diseases, including cancer, could enroll. Patients could not participate in pharmacological CTsor receive additional physiotherapy treatments while being enrolled and agreed to withdraw medication 12 h before blood draws.

All participants signed the corresponding informed consent previous admission. For symptom improvement monitoring and stability, the standardized validated instruments: FIQ [36,37], MFI [38] and the quality of life SF-36 questionnaire [39], in addition to a self-created CQ were used. Tender point sensitivities PPTs were measured before start (baseline) (pre-treatment), after completion of treatments (post-treatment) and one month after receiving the last treatment (post-washout). The latest was intended to estimate the washout period after MT effects (Figure 1). The washout period consisted in a 30-day waiting period immediately following the 8 sessions where patients agreed not to receive any physiotherapy treatment. The study was approved by the Universidad Católica de Valencia San Vicente Mártir Ethics Committee with study code UCV/2018-2019/076. All personal data were anonymized in fulfillment of Spanish data protection laws. Details of the trial are provided as a SPIRIT (Standard Protocol Items: Recommendations for Interventional Trials) checklist in compliance with interventional CT study design and registry good practice [43] (Supplementary File S1). All tasks were performed by collegiate professionals or qualified trained personnel. 

### 2.2. Intervention and Physiotherapist Self-Training

The customized intervention protocol consisted in basic massage maneuvers, including rubbing, friction and kneading, executed slowly, with wide glides, while applying gentle to moderate pressure (about 4.5 *n*) on patient’s back, 25/min duration each, 2 sessions weekly for a total of 8 in 4 consecutive weeks. The targeted force was considered enough to be effective while avoiding excessive muscular stress on the patient [31]. The treated muscles were the trapezius (upper, middle and lower fibers), supraspinatus and infraspinatus, rhomboids, teres major and minor, latissimus dorsi, lumbar square, paravertebral muscles, the entire thoraco-lumbar fascia, the pyramidal and the gluteal musculature (gluteus maximus, middle and minor). The protocol was applied on the patient on prone position with his/her back and gluteal areas exposed.

Prior to study start, the collegiate physiotherapist in charge of the CT self-trained to ensure reproducible pressure maneuvers of the desired intensity (4.5 *n*) by using a pressure-sensitive self-assembled artifact. The dispositive consisted in a flat soft-to-touch surface with a similar area to the body part to be treated, placed over four pressure sensors (Flexiforce, Tekscan, Canada). The training period consisted in repetitions of the protocol, until obtaining a triplicate of satisfactory pressure scores for 5 consecutive days. Only measures deviating 10% of the 4.5 *n* target force were considered satisfactory. Short pressure application operator self-checkups with the mentioned device preceded each patient session.

### 2.3. Questionnaires

In addition to the standard validated instruments to assess patient physical performance and emotional status, including the FIQ [36,37], MFI [38] and the SF-36 (Likert scale) [39] self-reported questionnaires, we developed an additional short tool to estimate patient’s perception of their own response to treatment (CQ). The items included 5 questions (Q1–Q5) scored with a scale of −1 to 1, with a −1 value if they felt worse, 0 if they felt no appreciable change or +1 if they sensed improvement right after receiving the treatment, or 24 h after. Appraisal of mobility, pain and fatigue was included in our CQ. In addition, an overall satisfaction question (Q6) using a scale of 1–10, with the highest value corresponding to highest level of satisfaction, was included to estimate patient’s overall assessment of the treatment. The formulated translated questions were:
Q1-How did you perceived the treated region right after each treatment?Q2-How did you perceived the treated region 24 h after each treatment?Q3-How is the mobility of your back after the full 8-session treatment?Q4-How is your back (sensation and pain) after the full 8-session treatment? (comparing before and after)Q5-How do you feel at the level of fatigue after the full 8-session treatment? (comparing before and after)Q6-How do you find the effectiveness of the treatment after its completion? (full 8-session treatment)

The CQ instrument was also used to evaluate patient’s perception on the stability of the treatment one month after the treatment ended. Q1-was reformulated to the following: How do you perceive the treated region now in comparison to the feeling right after treatment? Q2 was omitted, and Q6 was intended to assess the washout period for those finding the treatment effective (responders). Scores for the standardized instruments were calculated as previously described [38,39,40,41].

### 2.4. Pressure Point Threshold Assessment

To assess PPTs of tender point sites in patients with FM, we used a calibrated FDIX Force Gage, ForceOne algometer (Wagner Instruments, USA) with a capacity of 200 lbf/100 kgf/1000 N and with an accuracy of ± 0.2% dedicated (single force) and ± 0.3% interchangeable (multiple force) cell modules. A 1 cm^2^ rubber disk was used to press on the points to be evaluated using a 90° angle. The values were registered in lbf.

PPTs were measured for each of the 18 tender points located in the muscles/areas indicated for the diagnosis of FM. Bilaterally, they include occiput, trapezius, supraspinatus, gluteal, low cervical, second rib, lateral epicondyle humerus, greater trochanter and knee. The patient was placed in a supine position for the assessment of the following PPTs: low cervical, second rib, lateral epicondyle humerus, greater trochanter and knees; while for the assessment of the remaining PPTs (occiput, trapezius, supraspinatus and gluteal) the patient was placed in the prone position. The anatomic location of the pain points was appraised by the evaluating physiotherapist through palpation. The placement of the algometer was perpendicular to the point where the constant pressure was to be applied for sensitivity assessment. Each sampling consisted of 3 independent pressure measurements, allowing a recovery time of 10 s between each one. Before pressure tests started, patients were instructed to indicate when the pressure change caused pain, as previously described [44,45].

### 2.5. Blood Collection

A collection of blood fractions was generated and registered at the Institute of Health Carlos III National Biobank, Madrid, Spain, with the intention to evidence molecular changes induced by this pressure-controlled custom physiotherapy treatment and their stability. The collection consisted in plasma and PBMCs (peripheral blood mononuclear cells) obtained at baseline (before treatment started or pre-treatment), after the first therapeutic session, after the 4th therapeutic session, after completing the 8-session treatment (post-treatment) and after 30-day wait period (post-washout). Twenty ml aliquots were drawn by a collegiate nurse, using sodium citrate as anticoagulant (BD tubes, cat. 367691). Within 4 h after blood draw, the samples were centrifuged at 1500 rpm (453 g) during 15 min on a table preparative centrifuge. Plasma (approximate 8–12 mL) were transferred to 2-mL Eppendorf tubes and centrifuged to remove contaminating cells and platelets at 10,000× *g* for 10 min on a microcentrifuge. Supernatants were aliquoted into 1-mL cryovials. PBMCs were obtained from the rest of the blood, after removing the plasma, by dilution in phosphate-buffered saline solution (PBS) at 1:1 (*v*/*v*) ratio of the original volume, layering over one volume of Ficoll-Paque Premium (GE Healthcare, Chicago, IL, USA) followed by density centrifugation separation at 500 g for 30 min (20 °C, brakes off). The PBMC layer was washed with PBS and resuspended in red blood cell lysis buffer (155 mM NH4Cl, 10 mM NaHCO3, 0.1 mM EDTA, and pH 7.4), kept on ice for 5 min and centrifuged (20 °C at 500 g for 10 min), to remove contaminating erythrocytes. Washed pellets were snap frozen after removal of supernatant. Plasma aliquots were preserved at −80 °C and PBMCs at −150 °C until further use, as previously described [18,46].

Registration of the blood collection was done at the National Spanish Biobank Network (National Institute Carlos III) by completing the CT and collection details digitally on the Biobank Network website https://biobancos.isciii.es/NuevaColeccion.aspx


### 2.6. Statistical Analysis and Plotting

Continuous data are expressed as means ± SD and range as indicated. Statistical differences were determined using two-tailed unpaired *t*-tests for normal values and either Mann–Whitney or Wilcoxon nonparametric test if the values were found not to follow a normal distribution. Normal distribution of values was assessed by the Shapiro–Wilk normality test. Differences between groups were considered significant when *p* ≤ 0.05. Analysis were conducted with Excel and with the SPSS package 13.0 (SPSS Inc., Chicago, IL, USA) and R 3.6.1 [47]. Variable correlations were evaluated by the simple linear regression method (least-squares approach). Plots were drawn using the GraphPad Prism 5.0 program (San Diego, CA, USA) and the package ggplot2 [48].

## 3. Results

### 3.1. Study Design

The Clinical Trial NCT04174300 consisted in a pressure-controlled custom physiotherapy treatment of 10 out of the 18 FM tender points and surrounding areas towards triggering pain-reducing mechanisms, as outlined on Figure 1 and detailed on the Experimental Procedures section. Here we show the results for the outcome variables measuring pain, fatigue and quality of life, obtained with standard validated instruments and PPT scores, as an approach to evidence response to treatment. In addition, a CQ was added to the FIQ, MFI and SF-36 instruments [36,37,38,39] and algometer PPTs, which was used only after completing the 8-session treatment and after a 30-day washout period, consisting on a waiting time not receiving manual therapy to determine the stability of therapy-induced benefits. The trial included blood draws at different time points, as detailed in the Flowchart below and the Experimental Procedures section. The approximately 200 sample collection derived from 40 individuals donating blood at 5 time points (pre-treatment, after the first session, after completing the 4th session, after completing the 8-session procedure and 30-day after completing the 8-sessions as a waiting period without treatment (washout period)), yielded about 1600 aliquots (800 of plasma and 800 corresponding to PBMCs). This collection of samples was registered at the Institute of Health Carlos III National Biobank Network, Madrid, Spain with collection number C.0006173, with the intention to evidence molecular changes induced by this pressure-controlled custom physiotherapy treatment in future studies.

### 3.2. Cohort Demographics 

The final studied cohort included 38 FM patients (35 females and 3 males) who fulfilled 1990 and/or 2010 ACR criteria [2,3], with 50% (19/38) of them presenting comorbid ME/CFS according to Canadian and/or International diagnostic criteria [5,6]. Only one of the three participating males fulfilled ME/CFS criteria. Two of the initially 40 recruited patients had to be excluded for not completing the treatment, due to reasons unrelated to the therapy. Average age was 55.6 ± 7.2 years (range 43–71) and time from primary FM diagnosis over 3 years, 10.3 ± 7.5 years (range 3–21).

### 3.3. Patient Characteristics 

#### 3.3.1. Patient Health Status at Baseline

As determined with the indicated standard questionnaires (please see the Experimental Procedures section for details), total FM impact score of the studied cohort at baseline was 72.62 ± 15.67, range (41.08–96.26); general fatigue at the beginning of the study was 11.5 ± 1.62, range (7–16), which included physical and mental fatigue; and the perceived quality of life at recruitment was poor according to SF-36 scores, affecting physical, mental and emotional roles of participants (Table 1). According to the total FIQ scores of participants and taking into account that a score <39 indicates mild affection, ≥39 through ≤59 is assigned to moderately affected patients, and >59 to severe affection; the studied cohort can be described as mainly severe, with 79% (30/38) of the participants presenting severe FM and only 21% (8/38) moderate affection. Appendix A shows itemized findings.

#### 3.3.2. Blood Analytics of Participants

Blood analytics of participants did not show consistent alterations in any of the measured variables. However, it was interesting to observe that some patients presented off-range values for some parameters. In particular, it seemed noticeable that 63.8% (23/36) participants presented low percentage of neutrophils, according to normal range (Appendix A). Other blood parameters that deviated from established normal reference values were the percentage of monocytes, which appeared increased in 41.6% (15/36) of the study participants, and the high cholesterol (72.2% or 26/36) and low LDL (40% or 13/32). Mild hypercholesteremia in an advanced age group is considered usual and unrelated to the FM phenotype. An interesting less prevalent subgroup was that constituted by 6/36 participants (16.7%) with a higher than normal percentage of lymphocytes, which could be indicative of an infectious subgroup. However, only one of the 6 participants presented high absolute counts (Appendix A). A higher percentage of them (22.9% or 8/35) presented high levels of creatinine, which could relate to muscle damage. It should be noted that no blood analytics were available for 2 of the study participants and not all the variables were assayed for the complete group.

#### 3.3.3. Tender Point Sensitivity at Baseline

To estimate tender point sensitivity of participants (hyperalgesia/allodynia), each of the 18 anatomically sensitive areas described in FM [2] were localized and measured by triplicate with a FDIX Force Gage, ForceOne algometer (Wagner Instruments, USA), as detailed in Experimental Procedures. The average PPT values for each point, together with standard deviations and value range, are shown on Table 2. Scores show that the Low cervical points were the most sensitive while Gluteus, Trochanters and Knees were the most resistant to pain (Table 2). Appendix A show PPT average values for individual recordings. Ten out of the 18 PPTs were treated with our physiotherapy program while 8 of them were not, as indicated with asterisks on Table 2.

### 3.4. Patient Response to Therapy according to Questionnaires

The instruments used to evaluate pain, fatigue and quality of life at baseline (before the MT treatment started) were likewise used to assess changes in these parameters after completion of the 8-session treatment and also one month after the treatment ended, with the aim of estimating washout period of the response to treatment. As previously mentioned, patients agreed not to receive any physiotherapy session during the 30-day waiting period.

#### 3.4.1. Response to Treatment of FM Patients with or without Comorbid ME/CFS

As shown on Table 3, and detailed on Appendix A, only Total FIQ, FIQ Overall and SF-36 BP showed significant differences when scores after the physiotherapy treatment were compared to baselines (*p*-values (1)). Interestingly, differences for both FIQ total and FIQ Overall were also significant when the Pre-treatment and the Post-washout treatment were compared (*p*-values 2)), indicating that the benefits obtained from the therapy were lasting over a 30-day washout period. By contrast, SF-36 BP difference was not kept after this same wait period suggesting that although some general benefits persisted, the improvement in pain may be lost when therapy is discontinued for the indicated period of time.

When the criteria to assess the impact of the treatment on patient symptoms was our CQ, the results further supported that the 30-day wash period was enough to loose treatment benefits according to the significant differences found for the Response to treatment and the Total questionnaire scores, with lower means in the Post-washout (Table 4). Interestingly enough, patients perceived that the improvement or response to therapy was higher immediately after the treatment (0.97 ± 0.16) that 24 h after (0.39 ± 0.79). They also perceived no major benefits from the treatment to alleviate their fatigue.

#### 3.4.2. Response to Treatment of Patients with FM only vs. FM Patients with Comorbid ME/CFS

Since our cohort included 50% of the patients diagnosed for FM only, and 50% diagnosed with comorbid ME/CFS, and since manual therapy did not seem to affect the fatigue scores of patients (Table 3 and Table 4), we decided to explore whether the initial health status of the patient could play a role on the response to therapy. The argument being that the differential physiopathology attributed to these overlapping diseases [10] perhaps leads to different response in patients. For this purpose, we reassessed the statistical analysis performed after stratifying the participants in two groups (N = 19/each), according to having received diagnosis for one or both diseases by a collegiate clinician, as detailed in Experimental Procedures. 

It was interesting to find that several items from the questionnaires, particularly for the FIQ and the SF-36 questionnaires, clearly differentiated these subgroups of patients at all time points assessed (pre-, post- and post-washout), either with a tendency (*p* < 0.1), or with statistical significance (*p* ≤ 0.05). The increased significances observed for FIQ total and Overall observed (compare values of *p*-values 1 and 2 for these variables) seem to indicate preferential response of one of the two subgroups of patients being compared. This appears to be also the case for the SF-36 items: PF, VT, SF) and MH. Therefore, the benefits of the therapy seem to favor differently these two subgroups for pain and functioning, social activities and mental health (Table 5).

In fact, when the two subgroups of patients were individually analyzed for patient response to therapy (comparison at different time-points for each strata, or within group), the only significant differences that could be confirmed in the FM subgroup were the SF and MH variables, in addition to some tendencies for Total FIQ and Overall FIQ (Appendix A). Significant differences as measured by the SF-36 only affected the FM subgroup (Appendix A). By contrast, the other studied subgroup including patients with diagnosis of both FM&ME/CFS (*n* = 19) showed significant diferences for the FIQ Overall when the pre- and post-washout values were compared (p2, Appendix A), perhaps indicating a delayed benefit of the therapy. Interestingly, some statistical tendencies were found for MFI Physical fatigue and Reduced activity scores only in this last group (Appendix A), which could relate to the their ME/CFS diagnostic. The variability of the scores (SD values) and low number of participants after stratification (*n* = 19) may have hampered the identification of additional significant changes related to therapy response. Larger group analysis may be necessary to definitively determine symptom changes with therapy using these instruments.

Similarly, our CQ further supports preferential improvement of the FM subgroup with respect to the FM&ME/CFS (Table 6 and Table 7), highlighting pain response and mobility over fatigue.

### 3.5. Patient Response to Therapy according to PPTs

#### 3.5.1. Response to Treatment of FM Patients with or without Comorbid ME/CFS

As shown on Table 2, out of the five treated anatomic regions (Occiput, Trapezius, Supraspinatus, Gluteal, Low cervical) four tender points became less sensitive to pain pressure after the 8-session MT treatment, indicating overall patient’s improvement in the treated area. In particular, Trapezius and Gluteal significantly improved on both sides, while Occiput and Supraspinatus did not reach significance. Unexpectedly, the most sensitive to pain tender point among the treated, the Low cervical, increased pain sensitivity on both sides, suggesting a negative effect of the treatment in this region. PPTs of untreated areas, including the tender points, Second rib, Lateral epicondyle humerus, Greater trochanter and Knee, appeared unchanged (Table 2), except for of the right Second rib which sensitivity significantly increased after treatment. The significance of this change, if any, is presently unknown.

Interestingly enough, negative linear correlations between the initial PPT values and the differences or changes after treatment were observed for all 18 PPTs. This suggests that the initial status of the patient, with respect to pain perception, could be indicative of the patient’s capacity of response to the applied physiotherapy treatment. The findings show that in general patients with lower PPT scores (increased pain sensitivity) were capable of an increased benefit (highest difference or gained pain resistance after treatment, or change post-pre), while for those more resistant to pressure-induced pain (higher pre-treatment PPT values), therapy-induced benefits would be minor or absent (Figure 2).

Best R2 coefficients or goodness of the linear fit, on both body sides, corresponded to PPT values for the Low cervical within the sets of treated PPTs and to the Knee´s for the non-treated PPTs, as shown on Figure 2. These points at the sensitive Low cervical PPTs and the Knee PPTs may constitute potential predictors for the patient response to MT therapy. Moreover, the distribution of values shows that although mean values seem to indicate worsening of this highly sensitive tender point (Low cervical) by treatment, there is a subgroup with low PPT baseline values that obtain a benefit (Post-pre PPT difference value > 0 on Figure 2).

Individual plots for the assessment of PPT-dependent patient response for the remaining 14 tender points are available on Appendix A. Improved R2 values with respect to those of the Knee were obtained for the linear models of Second rib PPTs. However, the narrow distribution of the PPT values may lead to drastic changes in additional explorations, suggesting that the Knee PPTs provide a better option among the non-treated areas to predict patient’s response to therapy.

#### 3.5.2. Response to Treatment of Patients with FM only vs. FM Patients with Comorbid ME/CFS

Since patients diagnosed with FM only seemed to respond to physiotherapy more readily than patients with FM presenting comorbid ME/CFS (FM&ME/CFS), according to standard and our CQ questionnaires, it was of interest to reevaluate patients’ response to therapy by PPT assessment in either subgroup.

Interestingly enough, the results of these analysis show that opposite to the FM subgroup, PPTs were not good predictors of response to therapy in patients presenting comorbid ME/CFS (Figure 3 and Appendix A). In fact, patients with both, FM and ME/CFS showed poor or no response, independent of their sensitivity threshold to pain, for some tender points, as clearly shown on Figure 3B–D. Importantly, 5 out of the 10 PPTs treated showed significant differences between groups (*p* ≤ 0.05) (Figure 3B–E and Appendix A), and one additional (Supraspinatus right) showed a tendency (*p* < 0.1) (Appendix A), indicating differential response to treatment between groups. By contrast, none of the untreated PPTs showed any subgroup-associated difference (*p* > 0.1), except for the Second rib on the right side, which showed a tendency (*p* < 0.01) for differences to response between groups (Appendix A).

## 4. Discussion

The possibility of drug-free therapeutics seems highly attractive, particularly for diseases like FM and ME/CFS that appear often associated to varied comorbidities, multiple chemical hypersensitivity included [49]. Moreover, for the expected low undesired secondary effects and the avoidance of harmful interactions.

The rationale behind the proposal of using pressure-controlled MT to treat FM and ME/CFS is extensively presented and carefully discussed in a review paper published by our group about 2 years ago [30]. Some of the data supporting this proposal include several systematic reviews and meta-analysis of RCTs concluding that MT leads to beneficial effects on improving pain, anxiety and depression [28,30,50]. Another meta-analysis, including 140 studies has shown that MT reduces DOMS (delayed onset muscle soreness) and fatigue after exercise, more effectively than compression garment, electrostimulation, stretching, immersion or cryotherapy, as evidenced by decrease in the muscle damage marker creatine kinase (CK) and in the inflammation markers interleukin-6 (IL-6) and C-reactive protein [51], and therefore further supports MT as a potential therapy for FM.

MT activates mechano-transduction signaling pathways, induces mitochondria biogenesis signaling and diminishes the levels of inflammatory cytokines, such as the interleukin IL-6 and TNF (tumor necrosis factor)-α [52], changes that could benefit FM and CFS/ME patients [53,54]. Application of MT to soft and connective tissues leads to biochemical changes, at the time that allows local nociceptive and inflammatory mediators be reabsorbed by raising muscle blood and lymph circulation [55]. In fact, the use of pressure mimetic devices on animal models has shown that the intensity of therapeutic forces is determinant to obtain hypoalgesia effects [31,56]. Waters-Banker et al., by assaying low (1.4 N), medium (4.5 N) and high (11 N) loadings, found that high loadings drive to increased extracellular space, possibly as result of damage and edema of the muscle. While medium loading induced anti-inflammatory molecular changes and induced muscle regeneration, low pressures did not [31], indicative of the pertinence of a moderate controlled pressure MT protocol as the choice with therapeutic potential.

In compliance with these findings, some MT studies have strived to control the applied load, towards effect replicability, using approaches like patient pain feed-back scales, as for example includes the recently published work by Nadal-Nicolás et al. [57]. Since pain sensitivity varies across FM patients, we thought that pressure monitorization performed by a single pre-trained collegiate physiotherapist might increase inter-assay consistency. For this purpose, we developed a pressure-sensitive dispositive with built-in sensors that was used to ensure medium load MT protocols by self-training (please see Experimental Procedures for further details).

In line with our expectations, our medium load-pressure MT protocol consisting of 8-sessions of 25 min twice weekly for 4 consecutive days on alternate weekdays (Figure 1), led to improvement of patient pain thresholds, as measured by the standard FIQ and SF-36 questionnaires (Table 3). FIQ total and Overload were the most improved items, together with SF-36 Bodily pain. Similarly, CQ results showed most benefits for response immediately after treatment and upon completion of the whole set of 8-sessions (*p* < 0.05), in contrast to 24 h after treatment, when patients reported worse condition of the treated region (Table 4). It is possible that at that time (24 h post-treatment) some pressure-induced events lead to this feeling. It may be of interest to monitor patient evolution after treatment more closely in future trials. By contrast, no improvement was registered on patient’s fatigue, either by MFI or CQ scores (Table 3 and Table 4). As per the stability of the benefits obtained in pain reduction and overall status, we found that the effects persisted after a period free of treatment of 30-day (Table 3 and Table 4), at least to a certain extent. Longer patient follow ups are granted by continuation studies to determine MT effects washout period.

In agreement with the described questionnaire-assessed findings, differential PPTs measures obtained with a digital algometer, as described in Experimental Procedures, further confirmed significant improvement of patient’s hyperalgesia/allodynia in two of the treated sets of tender points, the Trapezius and the Gluteal. Only the Low cervical appear to worsen in the overall group assessment (Table 2). None of the untreated point PPTs were majorly affected with the only exception of the right Second rib, when the whole group was considered (*n* = 38), suggesting that the improvement was as an overall limited to the areas receiving the MT treatment. This disagrees with the compressing benefits reported by some authors, who evidenced systemic improvement on contralateral untreated limbs in experimental animal models [31,32]. However, when improvement PPT scores (difference between Post and Pre- PPT values: POST-PRE on Figure 2 and Appendix A), were plotted, we observed patients that were responders (POST-PRE values > 0) and others that appeared as not responders (POST-PRE values < 0), involving all, treated and untreated tender points (Figure 2 and Appendix A), indicating two different type of patients, and also evidencing the existence of systemic effects.

In reference to the appreciated effects on untreated areas, and its interpretation, we must point that a limitation of this study is the impossibility to attribute the applied mechanical loads to the effects obtained due to the absence of a sham treated arm (group of patients treated with a similar MT protocol using a low <1.4 N pressures) in our design (Figure 1). This arm was not included in this initial trial since our primary interest was to test the overall benefits and the performance of the output variables before investing any additional efforts. MT treatments inherently contain an emotional component, transferred to the patient through mental relaxation and stress reduction by the sense of touch. In line with this, several are the authors that have demonstrated beneficial health effects obtained from positive emotional stimulus, either reflected by the boosting of the immune system [34,35,58], and effects in weight gain and increased vagal tone [59], among others. Future interventions should add a sham group to discern mechanical from emotional triggered effects.

Unexpectedly, a negative correlation between patient’s initial hyperalgesia/allodynia status and MT therapeutic benefit was evidenced when the Change Post-Pre values were plotted as a function of their PPT baseline (Pre) values (Figure 2 and Appendix A). This importantly points out that patient sensitivity to pain, as assessed by its PPTs with an algometer, could serve as an indicator or predictor of the expected patient’s response to MT therapy. In particular, the data indicate that FM patients with worse health status (or at least with highest sensitivity to pain) will obtain most benefits while those presenting closer to normal pain threshold levels will not experiment improvements or will improve to a lesser extent. Among all 18 tender points assayed, we found that the Low cervical, in the treated group, and the Knee within the untreated, constitute the best sensors for this prediction, with prediction values of around 60%, reaching 74% individually (Figure 2 and Appendix A).

Moreover, we observed that while patients presenting only FM followed PPT improvement patterns inversely correlated with their baseline hyperalgesia/allodynia status, those diagnosed with comorbid ME/CFS did not respond to MT, at least for some of the treated points, as indicated by an overall lack of change in PPT values (Figure 3E, green lines). This suggests different responses for patients diagnosed with FM only with respect to those presenting comorbid ME/CFS, which is further supported by the obtained CQ scores (Table 6 and Table 7).

These observations open the unprecedented possibility of classifying patients as potential “responders” or “not responders” to pressure-controlled MT, based on their thresholds to pain and the presence of comorbid ME/CFS, further supporting somatic and mechanistic differences behind pain perception in one or the other disease. Nevertheless, since the number of observations here is limited, and the current trial is the first to detect these correlations, the findings will need to be validated in additional, more numerous cohorts.

In reference to the studied cohort, it should be stressed that the participating patients were mainly females (35/38), with an average age of 55.6 ± 7.2 years (range 43–71) and time from primary FM diagnosis over 3 years. Most (79%) presented severe affection while 21% corresponded to moderate cases. Physical problems overweighed emotional (Table 1). In addition, half of them had received diagnosis of ME/CFS. Potential immune and metabolic unbalance associated with low neutrophil percentages (63.8% of the participants), increased monocytes (41.6%), high lymphocytes (16.7%), high cholesterol (72.2%) and low LDL (40%) may be present. Increased levels of creatinine could relate to muscle damage in 22.9% of the participants (Appendix A, *n* = 36). The possibility that different cohorts of patients lead to distinct outcomes after applying this same protocol remains.

Differences between the participants having been diagnosed with FM only and those receiving diagnosis of both FM and ME/CFS at baseline were identified, mainly with the FIQ items Overall and Symptoms, and with the SF-36 Physical and Vitality items, MFI failed in detecting differences between these two groups. Similarly, FIQ and SF-36 were also superior at detecting differences between the response to MT of either group, with Overall and Total for the first questionnaire and Vitality, Social Functioning and Mental Health in the second. The differences found between groups 30-day after treatment withdrawal could relate to the mentioned observations of distinct response and the fact that the 30-day withdrawal seem not to be enough to washout treatment effects, mostly affecting again to physical roles, as assessed by the SF-36 instrument (Table 5). Thus, FIQ scores and SF-36 Vitality values may turn into additional indicators of the presence of ME/CFS comorbidity in FM.

Finally, the value of the collection of blood samples gathered and Biobanked in the context of this study results evident, as particular molecular profiles from some of the sets should permit revealing the cellular pathways mediating the response to this pressure controlled-MT protocol, as previously envisioned by our group [30]. Importantly differential molecular profiles will also allow us to understand the reasons behind patient’s response dependence according to patient’s hypoalgesia/allodynia status and with the co-diagnosis of ME/CFS. It is quite likely that these future studies yield new biomarkers for the prediction of response to MT, some perhaps allowing differential diagnosis of the FM and the ME/CFS disease.

## 5. Conclusions

In line with the main aims of this study:To determine the value of a custom pressure-controlled MT to alleviate FM symptoms as evidenced by scores of the standardized instruments FIQ [36,37], MFI [38] and SF-36 [39], plus our CQ;To improve patient hyperalgesia/allodynia, as determined by PPT increases of the 18 FM tender points [2], we find that, according to our predictions, our custom pressure-controlled MT protocol provides benefits for alleviating FM symptoms, as shown by FIQ and SF-36 scores, plus our CQ results, and that patient hyperalgesia/allodynia improves according to PPT values of the treated tender points.

Unexpectedly, we found that FM patients response to our custom MT treatment is dependent on the initial hyperalgesia/allodynia status of the patient, and on ME/CFS comorbidity, with improved outcomes for patients with increased pain sensitivity, as determined by PPT measurements, and for those presenting FM in the absence of concomitant ME/CFS.

Therefore, it is proposed that baseline PPT values and the diagnosis of concurrent ME/CFS could serve as criteria to predict patient response to pressure-controlled physiotherapy programs. In addition, standard FIQ and SF-36 itemized scores, individually or in conjunction with our CQ questionnaire, might help detect ME/CFS within FM cohorts, and perhaps serve to identify responders from those that do not respond to this type of mechanically triggered therapy.

Since a complete collection of blood samples is available from this study, it is expected that future molecular evaluations reveal the subjacent mechanisms to the identified patient response to MT treatment among others.

## Figures and Tables

**Figure 1 ijerph-17-08044-f001:**
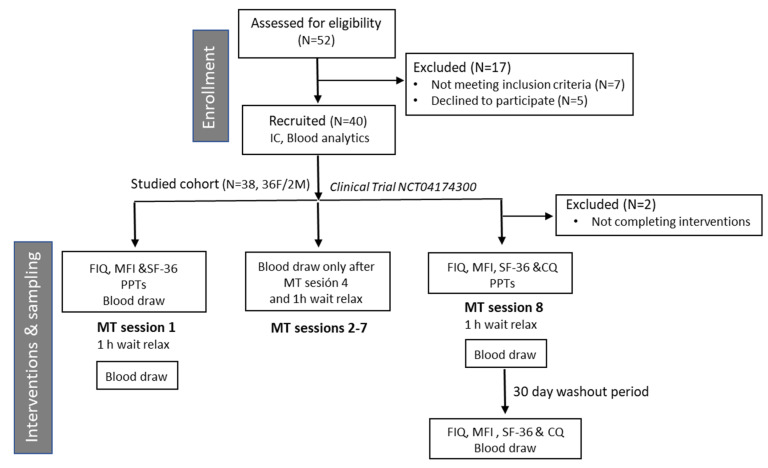
Flowchart depicting enrollment, intervention and sampling steps of the NCT04174300, following CONSORT (Consolidated Standards of Reporting Trials) guidelines [42]. Abbreviations: IC (Informed Consent), FIQ (Fibromyalgia Impact Questionnaire) [36,37], MFI (Multi Fatigue Inventory) [38], SF-36 quality of life questionnaire [39], CQ (Custom Questionnaire), PPTs (Pressure Point Threshold), MT (Manual Therapy).

**Figure 2 ijerph-17-08044-f002:**
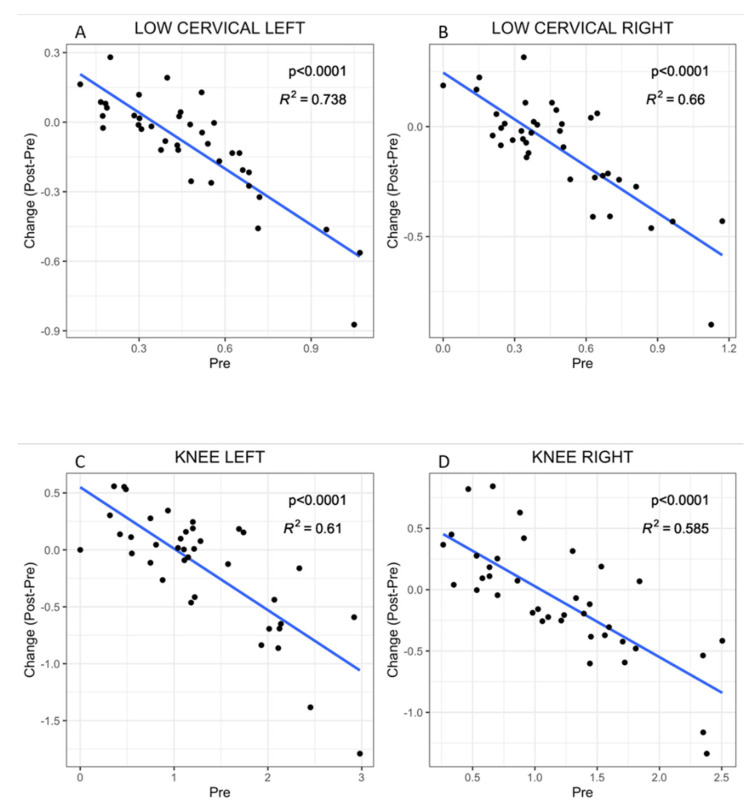
Linear dependence of PPT baseline scores with MT therapy response. The plots show the inverse correlation between pre-treatment PPT values and the improvement measured as the acquired resistance to pressure-induced pain for the tenderpoints (Change: Post–Pre) at the Low cervical left side of the body (**A**), or the right (**B**); and the same type of correlation for the tenderpoint at the left knee (**C**) or right knee (**D**). *p* and R2 values are shown.

**Figure 3 ijerph-17-08044-f003:**
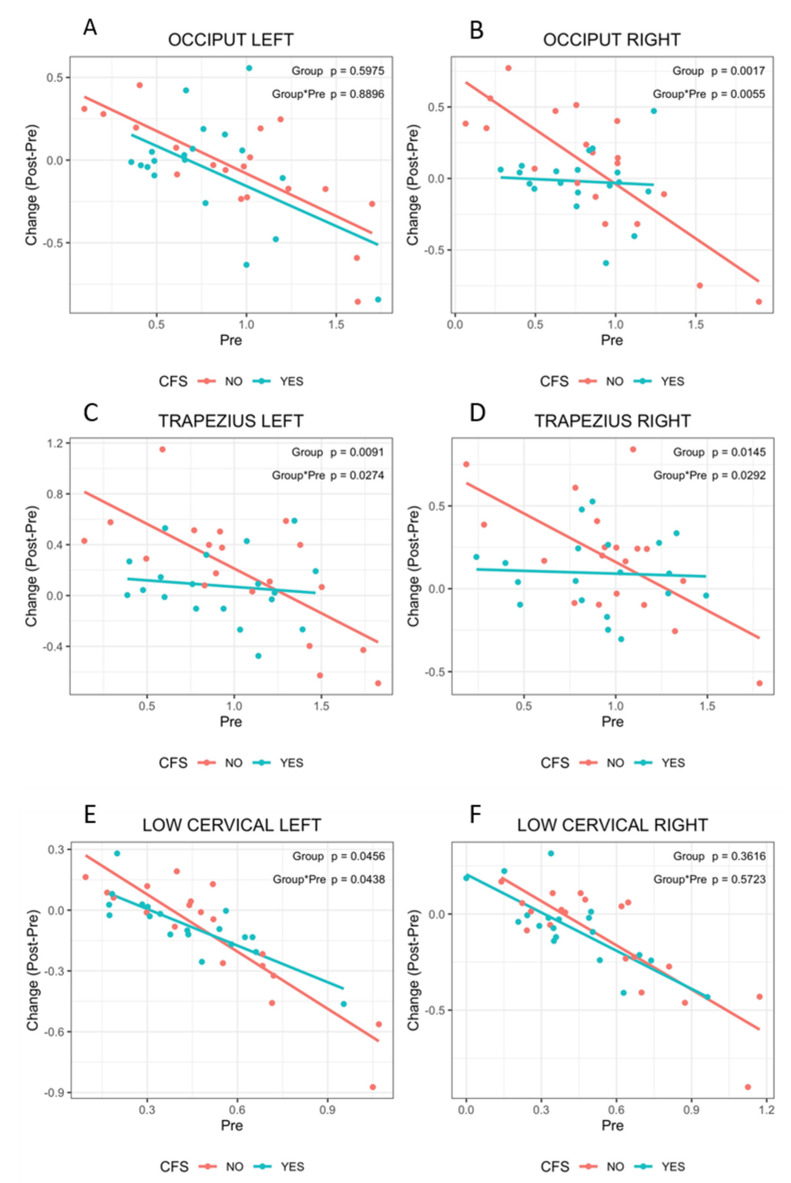
Linear dependence of PPT baseline scores (Pre) with MT therapy response (Change: Post–Pre) in patients presenting FM with or without comorbid ME/CFS. The plots show the inverse correlation between pre-treatment PPT values and symptom improvement, measured as the acquired resistance to pressure-induced pain for the tenderpoints at the Occiput left side of the body (**A**), or the right (**B**); and the same type of correlation for the tenderpoint at the left Trapezius (**C**) or its corresponding right PPT (**D**), and for the left Low cervical (**E**) or Low cervical right (**F**). Linear adjustments in the FM only subgroup are shown in red color, while those for the FM patients that present with ME/CFS comorbidity are shown in green. *p* values of the differences between FM only and FM with ME/CFS are shown.

**Table 1 ijerph-17-08044-t001:** Patient health status assessment with the FIQ, MFI and SF-36 instruments [36,37,38,39] at baseline (*n* = 38).

Questionnaire	Mean	SD	Range
**FIQ**			
Total FIQ	72.62	15.67	[41.08–96.26]
Function	5.16	2.29	[0–9.24]
Overall	8.3	2.29	[2.86–10.01]
Symptoms	4.59	2.23	[0–10.01]
**MFI**			
General Fatigue	11.5	1.62	[7–16]
Physical Fatigue	12.3	1.23	[10–16]
Reduced Activity	12.1	1.93	[6–19]
Reduced Motivation	10.6	2.72	[7–15]
Mental Fatigue	11.5	1.75	[7–15]
**SF-36**			
Physical Functioning (PF)	38.95	17.48	[0–85]
Role Physical (RP)	28.95	21.87	[0–81.25]
Bodily Pain (BP)	26.64	18.39	[0–70]
General Health (GH)	29.68	16.03	[0–65]
Vitality (VT)	16.12	15.83	[0–50]
Social Functioning (SF)	35.2	27.39	[0–87.5]
Role Emotional (RE)	56.58	37.12	[0–100]
Mental Health (MH)	47.24	21.92	[5–90]

FIQ (Fibromyalgia Impact Questionnaire) [36,37], MFI (Multi Fatigue Inventory) [38] and SF-36 quality of life questionnaire [39]. SD (standard deviation). Range refers to the possible values in the studied group.

**Table 2 ijerph-17-08044-t002:** Patient tender point sensitivity assessment, as determined by triplicate measurements in lbf with a FDIX Force Gage, ForceOne algometer (Wagner Instruments, Greenwich, CT, USA) at baseline (*n* = 38).

	Lbf (pound force)					
Tender Points	Mean PPTs pre-	Mean PPTs post-	SD pre-	SD post-	*p*-value	Range
Occiput. right *	0.8064	0.8404	0.3771	0.3187	0.6732	[0.065–1.525]
Occiput. left *	0.8714	0.8095	0.4029	0.3264	0.4646	[0.097–1.733]
Trapezius. right *	0.9341	1.0741	0.3521	0.3597	0.0903	[0.240–1.493]
Trapezius. left *	0.9725	1.1865	0.4149	0.6339	0.1890	[0.140–1.825]
Supraspinatus. right *	1.0115	1.0855	0.4311	0.4139	0.4478	[0.217–1.905]
Supraspinatus. left *	1.0050	1.0794	0.4045	0.3470	0.3924	[0.177–1.838]
Gluteal. right *	1.3286	1.5417	0.6430	0.5877	0.1359	[0.158–2.780]
Gluteal. left *	1.3550	1.5256	0.6481	0.6178	0.2443	[0.207–2.670]
Low cervical. right *	0.4865	0.3875	0.2684	0.1565	**0.0536**	[0.000–1.172]
Low cervical. left *	0.4774	0.3748	0.2347	0.1235	**0.0197**	[0.095–1.070]
Second rib. right	0.7117	0.5764	0.4746	0.2492	0.2971	[0.152–2.665]
Second rib. left	0.7098	0.6348	0.4530	0.2691	0.8424	[0.153–2.307]
Lateral Epicondyle humerus. right	0.8108	0.7335	0.4386	0.2781	0.3622	[0.080–1.830]
Lateral Epicondyle humerus. left	0.8495	0.7501	0.3602	0.3054	0.1305	[0.298–1.823]
Greater trochanter. right	1.9144	1.7825	0.9089	0.8721	0.3864	[0.285–3.892]
Greater trochanter. left	1.7636	1.8472	0.8524	0.8665	0.9034	[0.472–3.955]
Knee. right	1.1887	1.1096	0.6141	0.3921	0.5056	[0.263–2.505]
Knee. left	1.3025	1.1464	0.7296	0.4644	0.2608	[0.000–2.980]

(*) Tender points in areas treated with manual therapy; PPT (Pressure Point Threshold); SD (Standard Deviation); pre- (pre-treatment); post- (post-treatment). Significant differences (*p* ≤ 0.05) are bolded.

**Table 3 ijerph-17-08044-t003:** Patient response to therapy as evidenced by score differences in the standard, validated, FIQ, MFI and SF-36 instruments [36,37,38,39].

Questionnaire	Pre-Treatment	Post-Treatment	Post-Washout	*p*-value (1)	*p*-value (2)	*p*-value (3)
**FIQ**						
Total FIQ	72.62 ± 15.67	64.15 ± 18.25	63.82 ± 18.30	**0.0333**	**0.0274**	0.9370
Function	5.16 ± 2.29	4.62 ± 2.43	4.61 ± 2.48	0.3240	0.3209	0.9877
Overall	8.30 ± 2.23	6.74 ± 3.05	6.96 ± 2.79	**0.0122**	**0.0117**	0.8476
Symptoms	4.59 ± 3.72	4.14 ± 3.32	3.39 ± 3.14	0.7518	0.2139	0.2921
**MFI**						
General Fatigue	11.5 ± 1.6	11.7 ± 1.1	11.9 ± 1.7	0.6823	0.4383	0.3055
Physical Fatigue	12.3 ± 1.2	12.4 ± 1.8	12.4 ± 2.1	0.9092	0.8124	0.6592
Reduced Activity	12.1 ± 1.9	12.2 ± 2.3	12.4 ± 1.8	0.5709	0.8507	0.3838
Reduced Motivation	10.6 ± 2.7	10.7 ± 2.6	11.0 ± 2.2	0.3940	0.8765	0.3940
Mental Fatigue	11.5 ± 1.8	11.9 ± 1.7	11.6 ± 1.6	0.3972	0.4786	0.9058
**SF-36**						
Physical Functioning (PF)	38.95 ± 17.48	41.46 ± 16.55	41.71 ± 17.14	0.5218	0.9486	0.4887
Role Physical (RP)	28.95 ± 21.87	34.21 ± 25.24	32.40 ± 21.89	0.3419	0.7732	0.3441
Bodily Pain (BP)	26.64 ± 18.39	36.45 ± 23.65	32.11 ± 24.86	**0.0473**	0.2341	0.5879
General Health (GH)	29.68 ± 16.03	27.76 ± 15.14	27.11 ± 14.64	0.3763	0.9010	0.4654
Vitality (VT)	16.12 ± 15.83	20.53 ± 21.71	16.12 ± 16.09	0.6752	0.5948	0.9249
Social Functioning (SF)	35.20 ± 27.39	46.91 ± 27.20	45.72 ± 28.81	0.0563	0.8543	0.1004
Role Emotional (RE)	56.58 ± 37.12	53.51 ± 34.64	61.84 ± 35.49	0.6599	0.3037	0.5395
Mental Health (MH)	47.24 ± 21.92	54.08 ± 22.08	53.95 ± 24.47	0.1794	0.9804	0.2120

FIQ (Fibromyalgia Impact Questionnaire) [36,37], MFI (Multi Fatigue Inventory) [38] and SF-36 quality of life questionnaire [39]. SD (standard deviation). Range refers to the possible values in the studied group. *p*-value (1) refers to the pvalues obtained when the groups compared were the Pre- and the Post-treatment; *p*-value (2) refers to the pvalues obtained for comparison of the Pre- and the Post-washout sets of data; and *p*-value (3) to pvalues corresponding to differences between groups Post-treatment and Post-washout. Significant differences (*p* < 0.05) appear in bold.

**Table 4 ijerph-17-08044-t004:** Patient response to therapy assessed by score-differences in our CQ.

Custom Questionnaire	Post-Treatment	Range Post-Treatment	Post-Washout	Range Post-Washout	*p*-value
Response to treatment	0.97 ± 0.16	[0–1]	0.71 ± 0.61	[−1–1]	0.0197
Response to treatment after 24 h	0.39 ± 0.79	[−1–1]	N/A	N/A	N/A
Improves mobility	0.87C0.34	[0–1]	0.68 ± 0.47	[0–1]	0.0969
Pain reponse	0.82 ± 0.46	[−1–1]	0.66 ± 0.48	[0–1]	0.1106
Reponse to fatigue	0.24 ± 0.54	[−1–1]	0.24 ± 0.49	[−1–1]	>0.9999
Final treatment effectiveness	7.68 ± 2.44	[0–10]	6.88 ± 2.68	[0–10]	0.1558
Total questionnaire	10.97 ± 3.59	[−1–15]	9.17 ± 3.76	[0–14]	**0.0085**

N/A: not available. Significant differences (*p* < 0.05) appear in bold. Tendencies (*p* ≤ 0.1) are underlined.

**Table 5 ijerph-17-08044-t005:** Differences between FM patients presenting or not comorbid ME/CFS, assessed by FIQ, MFI and SF-36 instruments [36,37,38,39].

Questionnaire	Pre-Treatment FM&ME/CFS	Pre-Treatment FM	Post-Treatment FM&ME/CFS	Post-Treatment FM	Post-Washout FM&ME/CFS	Post-Washout FM	*p*-value (1)	*p*-value (2)	*p*-value (3)
**FIQ**									
Total FIQ	77.01 ± 13.66	68.22 ± 16.65	70.71 ± 13.87	57.60 ± 20.04	68.19 ± 19.60	59.45 ± 16.25	0.0836	**0.0247**	0.1436
Function	5.38 ± 1.85	4.93 ± 2.69	5.19 ± 2.24	4.05 ± 2.54	5.35 ± 2.15	3.87 ± 2.62	0.5509	0.1490	0.0654
Overall	9.08 ± 1.75	7.53 ± 2.42	7.60 ± 3.20	5.87 ± 2.69	7.38 ± 2.44	6.55 ± 3.10	**0.0249**	**0.0551**	0.5190
Symptoms	5.95 ± 3.35	3.24 ± 3.66	4.52 ± 3.17	3.76 ± 3.51	3.99 ± 3.89	2.78 ± 2.10	**0.0243**	0.3122	0.9945
**MFI**									
General Fatigue	11.63 ± 1.30	11.42 ± 1.92	11.74 ± 0.65	11.74 ± 1.37	12.00 ± 1.37	11.89 ± 2.08	0.2845	0.6913	0.9253
Physical Fatigue	12.16 ± 0.83	12.42 ± 1.54	12.00 ± 0.47	12.89 ± 2.40	12.95 ± 1.72	11.84 ± 2.34	0.7467	0.3336	0.1054
Reduced Activity	12.42 ± 1.98	11.68 ± 1.86	12.32 ± 2.11	12.16 ± 2.61	13.00 ± 1.83	11.74 ± 1.56	0.8644	0.9364	**0.0277**
Reduced Motivation	10.21 ± 2.35	11.00 ± 3.06	10.42 ± 2.61	10.95 ± 2.72	11.16 ± 2.32	10.84 ± 2.22	0.3777	0.5465	0.6703
Mental Fatigue	11.32 ± 1.70	11.63 ± 1.83	11.84 ± 1.54	12.00 ± 1.80	11.63 ± 1.12	11.53 ± 1.93	0.5854	0.7504	0.8378
**SF-36**									
Physical Functioning (PF)	34.47 ± 16.49	43.42 ± 17.72	36.84 ± 25.29	46.08 ± 16.85	36.32 ± 18.02	47.11 ± 14.75	0.1159	0.0853	**0.0509**
Role Physical (RP)	22.37 ± 19.91	35.53 ± 22.25	28.29 ± 25.55	40.13 ± 24.14	26.32 ± 21.51	38.49 ± 21.07	**0.0319**	0.1538	0.0865
Bodily Pain (BP)	20.79 ± 14.12	32.50 ± 20.58	29.74 ± 18.26	43.16 ± 26.86	26.71 ± 23.86	37.50 ± 25.29	0.0654	0.0800	0.1715
General Health (GH)	29.47 ± 17.07	29.89 ± 15.39	25.79 ± 15.12	29.74 ± 15.32	26.84 ± 18.57	27.37 ± 9.77	0.9368	0.5171	0.7656
Vitality (VT)	7.89 ± 10.59	24.34 ± 16.12	7.89 ± 9.96	33.16 ± 23.08	8.88 ± 12.20	23.36 ± 16.52	**0.0018**	**<0.0001**	**0.0019**
Social Functioning (SF)	27.63 ± 28.13	42.76 ± 25.11	38.16 ± 27.15	55.66 ± 24.94	30.92 ± 25.13	60.53 ± 24.74	0.0597	**0.0458**	**0.0007**
Role Emotional (RE)	50.00 ± 39.28	63.16 ± 34.62	46.49 ± 33.25	60.53 ± 35.45	52.19 ± 39.37	71.49 ± 29.04	0.3306	0.2001	0.1887
Mental Health (MH)	43.95 ± 24.58	50.53 ± 19.00	46.48 ± 24.62	61.32 ± 16.90	45.26 ± 28.31	62.63 ± 16.45	0.3621	**0.0267**	**0.0266**

FIQ (Fibromyalgia Impact Questionnaire) [36,37], MFI (Multi Fatigue Inventory) [38] and SF-36 quality of life questionnaire [39]. FM (Fibromyalgia only), FM&ME/CFS (Fibromyalgia with comorbid ME/CFS). *p*-value (1) refers to the pvalues obtained when the groups compared were the Pre-treatment FM&ME/CFS and the Pre-treatment FM; *p*-value (2) refers to the pvalues obtained for comparison of the Post-treatment FM&ME/CFS and the Post-treatment FM sets of data; and *p*-value (3) to pvalues corresponding to differences between groups Post-washout FM&ME/CFS and the Post-washout FM. Significant differences (*p* ≤ 0.05) appear in bold. Tendencies (*p* < 0.1) are underlined.

**Table 6 ijerph-17-08044-t006:** FM only patient subgroup response to therapy, assessed by score-differences in our CQ.

Questionnaire	Post-Treatment	Range Post-Treatment	Post-Washout	Range Post-Washout	*p*-value
Response to treatment	1.00 ± 0.00	[1–1]	0.79 ± 0.54	[−1–1]	0.2297
Response to treatment after 24 h	0.42 ± 0.84	[−1–1]	NA	NA	NA
Improves mobility	1.00 ± 0.00	[1–1]	0.74 ± 0.45	[0–1]	0.0463
Pain reponse	0.95 ± 0.23	[0–1]	0.58 ± 0.51	[0–1]	0.0188
Reponse to fatigue	0.37 ± 0.60	[−1–1]	0.37 ± 0.60	[−1–1]	>0.9999
Final treatment effectiveness	8.19 ± 1.77	[3–10]	7.26 ± 2.05	[3–10]	0.1814
Total questionnaire	11.89 ± 2.38	[5–15]	9.74 ± 3.09	[3–14]	0.0256

N/A: not available. Significant differences (*p* < 0.05) appear in bold.

**Table 7 ijerph-17-08044-t007:** FM&ME/CFS patient subgroup response to therapy, assessed by score-differences in our CQ.

Questionnaire	Post-treatment	Range Post-treatment	Post-washout	Range Post-washout	*p*-value
Response to treatment	0.95 ± 0.23	[0–1]	0.63 ± 0.68	[−1–1]	0.1264
Response to treatment after 24 h	0.37 ± 0.76	[−1–1]	NA	NA	NA
Improves mobility	0.74 ± 0.45	[0–1]	0.63 ± 0.50	[0–1]	0.7281
Pain reponse	0.68 ± 0.58	[−1–1]	0.74 ± 0.45	[0–1]	>0.9999
Reponse to fatigue	0.11 ± 0.46	[−1–1]	0.11 ± 0.32	[0–1]	>0.9999
Final treatment effectiveness	7.21 ± 2.94	[0–10]	6.50 ± 3.21	[0–10]	0.5045
Total questionnaire	10.05 ± 4.36	[−1–14]	8.61 ± 4.34	[0–13]	0.2275

N/A: not available. Significant differences (*p* < 0.05) appear in bold.

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
