# Peer review of "Pressure Point Thresholds and ME/CFS Comorbidity as Indicators of Patient’s Response to Manual Physiotherapy in Fibromyalgia"

_ijerph, 2020, doi:10.3390/ijerph17218044_

Round 1
Reviewer 1 Report
It might be interest that the conclusions can appear clearer, showing the answers to the objetive set (aims).
Separate and specify the objectives separately, is recommended.
Table 5: It`s a bit difficult to read this table. It would be necessarily to improve for can read the results between two groups depending on each questionnaire. For example: In this table: p=0,0836 ; p-(value 1: pre and de post treatment), now, Which group does this value belong to?, between groups?, within group?
Table 5: I can not interpet the significance value that belongs to each group. I think that it will be necessary to improve the presentation of this data to know what value is associated with each group based on the different questionnaires.
Author Response
Reviewer 1: Comments and Suggestions for Authors
It might be interest that the conclusions can appear clearer, showing the answers to the objetive set (aims).
Separate and specify the objectives separately, is recommended.
Thanks for the suggestion. For further clarity, we have now listed the initial aims of the study at the start of the Conclusions section:
- “determine the value of a custom pressure-controlled MT to alleviate FM symptoms as evidenced by scores of the standardized instruments FIQ [38,39], MFI [40] and SF-36 [41], plus our CQ;
- and to improve patient hyperalgesia/allodynia, as determined by PPT increases of the 18 FM tender points [2], “
followed by the answers:
“..we find that, according to our predictions, our custom pressure-controlled MT protocol provides benefits for alleviating FM symptoms, as shown by FIQ and SF-36 scores, plus our CQ results; and that patient hyperalgesia/allodynia improves according to PPT values of the treated tender points”
Table 5: It`s a bit difficult to read this table. It would be necessarily to improve for can read the results between two groups depending on each questionnaire. For example: In this table: p=0,0836 ; p-(value 1: pre and de post treatment), now, Which group does this value belong to?, between groups?, within group?
P-value (1) on Table 5 refers as indicated in the corresponding table foot annotation to: “…pvalues obtained when the groups compared were the Pre-treatment FM&ME/CFS and the Pre-treatment FM…”. So, in this case the comparison was made at one time point (Pre-treatment) between groups: the subgroup of patients diagnosed with both FM & ME/CFS and the subgroup of patients diagnosed of FM only.
Table 5: I can not interpet the significance value that belongs to each group. I think that it will be necessary to improve the presentation of this data to know what value is associated with each group based on the different questionnaires.
In line with the response to the previous comment we have accommodated the introductory paragraph of the 3.4.2 section to guide the reader into the significance of patients being stratified according to being diagnosed with one (only FM) or both (FM & ME/CFS) diseases. We have also completely rewritten the description of the findings to increase clarity about what it is being compared in the different tables.
Table 5-comparison between groups (FM vs FM &ME/CFS)
Supplementary Tables 4 and 5- comparisons within groups of patients, diagnosed only with FM or with both, respectively.
Reviewer 2 Report
It's a good idea to uses scores from validated standardized questionnaires, algometer pressure point threshold readings and responses from a custom self-developed questionnaire to determine the impact of a pressure-controlled manual protocol on Fibromyalgia hyperalgesia/allodynia, fatigue and patient´s quality of life. A systematically evaluation of physiotherapy treatments or in combination with current pharmacological treatments is very useful to determine its impact on Fibromyalgia hyperalgesia/allodynia, fatigue and patient’s quality of life.
However, regarding to the prevention of liver fibrosis, the underlying mechanism still not clear. I would like to ask few questions, which would be great if it can be addressed.
- It would be great if the authors could give some details regarding to patients’ general information, such as other related medical history.
- There is deviation of gender, it would be nice if authors could give some discussion about the difference or potential effects on the data interpretation.
Author Response
Reviewer 2: Comments and Suggestions for Authors
It's a good idea to uses scores from validated standardized questionnaires, algometer pressure point threshold readings and responses from a custom self-developed questionnaire to determine the impact of a pressure-controlled manual protocol on Fibromyalgia hyperalgesia/allodynia, fatigue and patient´s quality of life. A systematically evaluation of physiotherapy treatments or in combination with current pharmacological treatments is very useful to determine its impact on Fibromyalgia hyperalgesia/allodynia, fatigue and patient’s quality of life.
Thank you for the comment.
However, regarding to the prevention of liver fibrosis, the underlying mechanism still not clear. I would like to ask few questions, which would be great if it can be addressed.
- It would be great if the authors could give some details regarding to patients’ general information, such as other related medical history.
All the medical history available for the participants has been summarized in the manuscript or manuscript supplementary files.
- There is deviation of gender, it would be nice if authors could give some discussion about the difference or potential effects on the data interpretation.
This is a very interesting point as gender differences are starting to be evidenced by molecular marker analysis. For example, our group recent publication:
Cheema AK, Sarria L, Bekheit M, Collado F, Almenar-Pérez E, Martín-Martínez E, Alegre J, Castro-Marrero J, Fletcher MA, Klimas NG, Oltra E, Nathanson L. Unravelling myalgic encephalomyelitis/chronic fatigue syndrome (ME/CFS): Gender-specific changes in the microRNA expression profiling in ME/CFS. J Cell Mol Med. 2020 May;24(10):5865-5877. doi: 10.1111/jcmm.15260
However, the low number of male participants in the present study limits stratification of data by gender. It will be interesting to design future physiotherapy studies that include larger groups of both male and female patients.
Reviewer 3 Report
present in the summary the objective and methodology of the study more clearly and thus, an objective conclusion in response to its goal.
inform in the methodology how the questionnaire scores are calculated.
inform if the algometer was calibrated.
inform if the person responsible for each stage of the method has been trained for that specific procedure.
Author Response
Reviewer 3: Comments and Suggestions for Authors
present in the summary the objective and methodology of the study more clearly and thus, an objective conclusion in response to its goal.
In the abstract, the sentence:
“This study uses scores from validated standardized questionnaires, algometer pressure point threshold (PPT) readings and responses from a custom self-developed questionnaire to determine the impact of a pressure-controlled manual protocol on FM hyperalgesia/allodynia, fatigue and patient´s quality of life”
Already includes methodology (underlined above) and objective (bold above).
In addition, we have now added the following text in the Conclusion section to clarify correlations between the initial aims set and the unexpected findings, some deriving from post-stratification analysis:
“In line with the main aims of this study:
- determine the value of a custom pressure-controlled MT to alleviate FM symptoms as evidenced by scores of the standardized instruments FIQ [38,39], MFI [40] and SF-36 [41], plus our CQ;
- and to improve patient hyperalgesia/allodynia, as determined by PPT increases of the 18 FM tender points [2],
we find that, according to our predictions, our custom pressure-controlled MT protocol provides benefits for alleviating FM symptoms, as shown by FIQ and SF-36 scores, plus our CQ results; and that patient hyperalgesia/allodynia improves according to PPT values of the treated tender points”
inform in the methodology how the questionnaire scores are calculated.
A sentence has been added to the section 2.3, as follows:
“Scores for the standardized instruments were calculated as previously described [38-41].”
inform if the algometer was calibrated.
This information has now been added to section 2.4, as follows:
“..we used a calibrated FDIX Force Gage, ForceOne algometer”
inform if the person responsible for each stage of the method has been trained for that specific procedure.
This information has now been added at the end of section 2.1, as follows:
“All tasks were performed by collegiate professionals or qualified trained personnel.”
Additional information is available in the last paragraph of section 2.2, as follows:
“Prior to study start the collegiate physiotherapist in charge of the CT self-trained…”
To clarify that the physiotherapist responsible for the MT was a collegiate professional self-trained to reproduce a constant application of pressures, using indicators sensitive to pressure that objectively register force units.